# Experimental Validation of a Thermo-Electric Model of the Photovoltaic Module under Outdoor Conditions

**Klemen Sredenšek** [1,*] , **Bojan Štumberger** [1,2], **Miralem Hadžiselimović** [1,2], **Sebastijan Seme** [1,2] **and Klemen Deželak** [1]

1   Faculty of Energy Technology, University of Maribor, Hočevarjev trg 1, 8270 Krško, Slovenia; bojan.stumberger@um.si (B.Š.); miralem.h@um.si (M.H.); sebastijan.seme@um.si (S.S.); klemen.dezelak@guest.um.si (K.D.)
2   Faculty of Electrical Engineering and Computer Science, University of Maribor, Koroška cesta 46, 2000 Maribor, Slovenia
*   Correspondence: klemen.sredensek@um.si

**Abstract:** An operating temperature of the photovoltaic (PV) module greatly affects performance and its lifetime. Therefore, it is essential to evaluate operating temperature of the photovoltaic module in different weather conditions and how it affects its performance. The primary objective of this paper is to present a dynamic thermo-electric model for determining the temperature and output power of the photovoltaic module. The presented model is validated with field measurement at the Institute of Energy Technology, Faculty of Energy Technology, University of Maribor, Slovenia. The presented model was compared with other models in different weather conditions, such as clear, cloudy and overcast. The evaluation was performed for the operating temperature and output power of the photovoltaic module using Root-Mean-Square-Error (RMSE) and Mean-Absolute-Error (MAE). The average RMSE and MAE values are 1.75 °C and 1.14 °C for the thermal part and 20.34 W and 10.97 W for the electrical part.

**Keywords:** dynamic modeling; thermo-electric model; accuracy; measuring device; temperature; output power; PV module

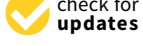

## 1. Introduction

The commercial use of photovoltaic modules dates back to the 1970s and 1980s of the 20th century. Ease of use, maintenance and installation led to mass integration in the early 21st century. The PV systems are classified into grid-connected and off-grid applications according to the type of grid connection [1]. Nowadays, due to self-sufficiency, the off-grid system is a very common way of connection, as it presents a great energy potential in underdeveloped countries or in countries without a well-established electricity network. To facilitate the digital transformation of energy infrastructure, it is also necessary to encourage the development and introduction of smart grids [2–4] and microgrids [5]. A crucial factor for PV system integration is also the assessment of energy potential [6] at a specific location, which ranges from 650 to 1500 kWh/kW$_p$ for the area of Europe.

Electricity production of PV systems depends on parameters, such as solar radiation, operating temperature of the PV module and air mass factor. Nevertheless, it is extremely important to assess the performance of the PV system in partial shading conditions [7]. The operating temperature of the PV module mainly affects the electrical parameters of the PV cell, such as saturation current, photon current, shunt and series resistance, diode ideality factor, short-circuit current, open-circuit voltage and consequently, the output power of the PV module [8]. An increase in temperature of the PV module by 1 °C results in a 0.45% drop in the output power [9] or a 0.22% drop in efficiency of the PV module [10]. The temperature under standard test conditions (STC) is defined as the operating temperature of the PV module, but only in the PV cell layer. It is very difficult or almost impossible to measure

the temperature of the PV cell layer in various outdoor conditions. Therefore, various models have been developed in order to determine the operating temperature of PV cells with help of meteorological and experimental data [11–15]. In addition to meteorological data, operating temperature of the PV module is also influenced by the type of materials and operation PowerPoint at which the PV module operates. The operating temperature of the PV cell is higher than the temperature of the backside or backsheet layer. However, the measurements of the operating temperature of the PV cell are almost impossible to perform under outdoor conditions. Therefore, the temperature sensors and all other measuring equipment should be calibrated in an estimated time to avoid some unnecessary errors. The most common equipment used for meteorological measurements is the platinum resistance thermometer (temperature sensor), anemometer (wind speed and direction sensor) and pyranometer (solar irradiance). Pyranometer [16] and anemometer [17] are some of the most sensible sensors that need to be calibrated in the estimated time (approx. 1–2 years). Otherwise, they can cause measurement uncertainties that increase with age.

### 1.1. Literature Review of the Existing Studies

Based on a large number of different models for estimating the temperature distribution in the PV module, a comprehensive review of existing studies was performed in this part of the study. Many static and dynamic models have been developed to estimate the temperature distribution in the PV module accurately. Since temperature distribution in the PV module is described differently for many types and designs, Ross [18] provides an overview of design requirements, design analysis and test methods for flat-plate PV modules. At that time, the NOCT model was usually used for temperature assessment of the PV module. Similarly, Faiman [19] presents the flat-plate PV module outdoor operating temperature assessment using a simple modified form of the Hottel-Whillier-Bliss (HWB) equation. Furthermore, Sandia National Laboratories [20] proposed a new empirically-based thermal model for flat-plate PV modules mounted in an open rack. The thermal model consists of temperature calculation of the backsheet layer and PV cell. This model has proven to be adaptable and adequate for flat-plate PV modules with an accuracy of $\pm 5$ °C. The results show that the temperature of the PV cell and the temperature of the backsheet layer can differ significantly, especially for different types of PV modules. In their study, Migliorini et al. [21], review two different approaches for determining a physical model of the PV module. The first model consists of a detailed temperature description, while the second electric model, is described by an explicit equation. The second model consists of a detailed electric model for I-V curve prediction, while the NOCT model describes the operating temperature of the PV module. Li et al. [10] propose a comprehensive multi-state dynamic thermal model for temperature estimation of the PV cell. Temperatures are modeled as internal states and are corrected according to the observations, which leads to high accuracy of the proposed model and reliable prediction of PV cells temperature and output power. Barry et al. [22] present a simple dynamic thermal model using non-linear optimization with four parameters. The proposed model reduces RMSE between measured and modeled PV module temperature to 1.58 °C on average. However, it does not consider optical losses and losses due to power generation but uses both global and diffuse solar radiation as an input parameter. In their study, Yu et al. [23], focus on the 2D temperature-irradiance coupling model of the flat-plate photovoltaic/thermal (PV/T) collector, which is very similar to the described model in [21], without the energy balance equation of water flow in copper tubes. This model includes the global and diffuse solar radiation as its input and considers the change of the absorptivity, transmissivity and reflectivity at a specific incident angle, which can be calculated by Snell's law.

### 1.2. Aims and Specifics of the Current Research

The originality of the current study lies in the development of a thermo-electric model for the assessment of temperature distribution in PV module. In this paper both, thermal and electric models of the PV module are described with dynamic modeling. The dynamic

thermal model is used from [10]. While the dynamic electric model is presented by a double diode PV cell model, including electrical parameters as a function of solar radiation and the operating temperature of the PV module. Modeling of the thermo-electric model of the PV module was performed in Matlab/Simulink software package using the s-function block diagram. Validation of the presented thermo-electric model is based on meteorological and experimental data of the PV system used in this study. By accurately determining the temperature distribution in the PV module, also lifetime of the PV module can be predicted. In addition, two other models for determining the temperature distribution in the PV module and the output power are included. The temperature distribution in the PV module was calculated using the Ansys Transient Thermal software package, and the already confirmed empirical equation [10,24–29] was used to calculate the output power of the PV module. In addition to validating the presented dynamic thermo-electric model, the importance of accuracy of measuring devices and optical losses is also included. Various authors [10,12,13,23,30] used constant values of optical losses in their models to determine the temperature distribution of fixed PV modules. Even though, in this paper measurements are performed on a dual-axis tracking system, the optical losses were calculated as a function of an incident angle of the Sun's rays, which is similar as in the following studies [31,32]).

The paper consists of five sections. The first section provides an introduction and literature review. The second section presents the methodology and measurement site, the third section includes some results of the proposed model and the fourth section presents thediscussion of the results compared to other studies. The last section presents the essential findings or conclusions.

## 2. Materials and Methods

Our study covers more than 35 scientific papers, published from 1982 to 2021, in which authors deal with different approaches to determine the temperature distribution on the backside of the PV module and/or the output power of the PV module. The first subsection presents the experimental set-up on where the measurements were performed, while the second and third subsections describe the mathematical equations for defining the thermal and electric parts of the PV module.

### 2.1. Experimental Set-Up

The measurements required for the validation of thermo-electric models were performed in the experimental field of the Institute of Energy Technology, Faculty of Energy Technology, University of Maribor. The experimental field consists of nine dual-axis PV tracking systems with additional measuring equipment (shown in Figure 1). One dual-axis PV tracking system consists of twenty series-connected 260 $W_p$ silicon PV modules with a total installed power of 5.2 $kW_p$. Each PV tracking system is equipped with a pyranometer for solar radiation measuring, four temperature sensors for measuring the operating temperature of the PV module on the backside and a DC/AC inverter. The DC/AC inverter does not include measurements of AC and DC current, voltage and output power data, therefore a Power Meter (Siemens SENTROM PAC4200—AC measurements) and a Hall Effect Sensor (T201DCH100—DC measurements) are additionally installed. The entire system is connected to a several-local-and-one-global PLC (Siemens S7-300), while the data is displayed in a SCADA software system that allows easy monitoring and management. All meteorological and experimental data are sampled every 5 min. Several detailed information on the accuracy and calibration time of the measuring devices are shown in Table 1.

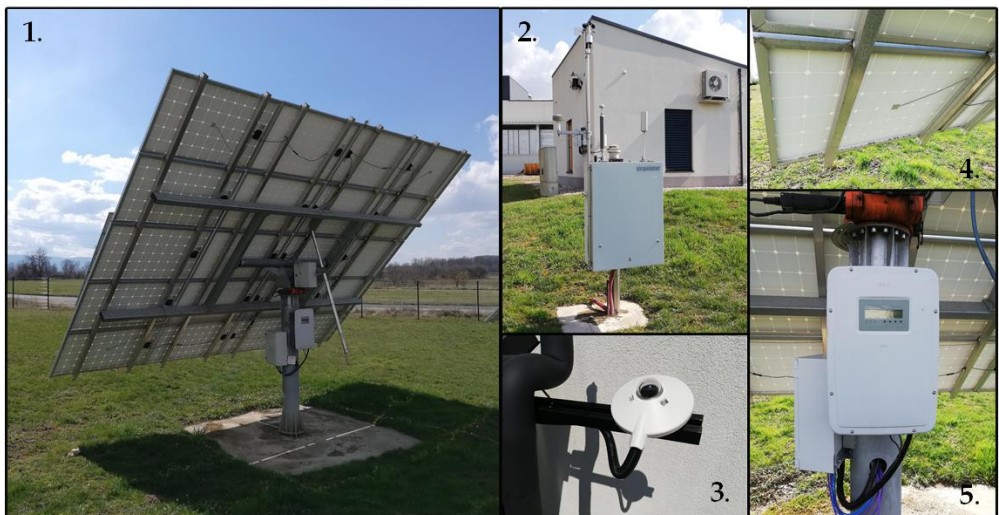

**Figure 1.** Measurement and experimental system: (**1**) Dual-axis PV tracking system; (**2**) Ambient temperature sensor; (**3**) Solar radiation sensor—pyranometer; (**4**) Temperature sensor; and (**5**) DC/AC inverter. @Laboratory for Applied Electrical Engineering, Faculty of Energy Technology, University of Maribor.

**Table 1.** Characteristics of measurement devices.

|  | **Uncertainty** | **Calibration** |
|---|---|---|
| Temperature sensor: DS18x20 | $\pm 0.5\,^\circ$C (from $-10$ to $85\,^\circ$C) | 6 years ago |
| Wind speed sensor: VMT107A | $\pm 0.5$ m/s | 6 years ago |
| Solar radiation sensor—pyranometer: Kipp&Zonen SP Lite 2 | <1% | 6 years ago |
| Ambient temperature sensor: TPR 159 (Pt100) | $\pm 0.15\,^\circ$C | 6 years ago |
| AC measurement—Power Meter: Siemens Sentron PAC4200 | Current: $\pm 0.2\%$ Voltage: $\pm 0.2\%$ | 3 months ago |
| DC measurement—Hall Effect Sensor: T201DCH100 | Current: $\pm 0.5\%$ Voltage: $\pm 0.5\%$ | 3 months ago |

*2.2. Dynamic Electric Model of the Photovoltaic Module*

Electric models of the PV module could be found in several papers [10,29,33] as a simple empirical correlation between output power $P_{DC}$, solar radiation $G$ and operating temperature of the PV module $T$ or as a single/double diode model [34–38] using the algorithm for a maximum power point tracking (MPPT). In this paper, the dynamic electric model is presented as a double diode model taking into account four parameters (short-circuit current $I_{SC}$, open-circuit voltage $V_{OC}$, series resistance $R_s$ and shunt resistance $R_{sh}$) as a function of solar radiation $G$ and the operating temperature of the PV module $T$. The double diode model of the PV cell is expressed by (1):

$$I = I_{ph} - I_{01} \cdot \left( \exp^{\left(\frac{V+I\cdot R_s}{V_{T1}\cdot N_s}\right)} - 1 \right) - I_{02} \cdot \left( \exp^{\left(\frac{V+I\cdot R_s}{V_{T2}\cdot N_s}\right)} - 1 \right) - \left( \frac{V+I\cdot R_s}{R_{sh}} \right) \tag{1}$$

where $N_s$ is the number of series-connected PV cells, $I_{ph}$ is the photocurrent, $I_{01}$ and $I_{02}$ are the reverse saturation current of the first and second diode, while $V_{T1}$ and $V_{T2}$ stand for thermal voltage of the first and second diode. The last five parameters ($I_{ph}$, $I_{01}$, $I_{02}$, $V_{T1}$ and $V_{T2}$) are described in more detail in Appendix A. As mentioned above, a short-circuit current, an open-circuit voltage, a series and shunt resistance [39] of the PV module are calculated as a function of the solar radiation $G$ and the operating temperature of the PV module $T$ by (2)–(5), where $K$ is the Boltzman

n constant ($1.38065 \times 10^{23}$ J/K), $q$ is the electron charge ($1.602 \times 10^{19}$ C) and $n$ is the diode ideality factor (dimensionless). Furthermore, $\mu_{I_{SC}}$ and $\mu_{V_{OC}}$ are temperature coefficients of $I_{SC}$ and $V_{OC}$, while $I_{SC,STC}$, $V_{OC,STC}$, $G_{STC}$ and $T_{STC}$ presents short-circuit current, open-circuit voltage, solar radiation and temperature of the PV module under STC conditions.

$$I_{SC}(G,T) = \left( \frac{G}{G_{STC}} \right)^{\frac{\ln\left(\frac{I_{SC,STC}}{I_{SC}}\right)}{\ln\left(\frac{G_{STC}}{G}\right)}} \cdot \left( I_{SC,STC} + \mu_{I_{SC}} \cdot (T - T_{STC}) \right) \tag{2}$$

$$V_{OC}(G,T) = V_{OC,STC} + \frac{N_S \cdot K \cdot T \cdot n}{q} \cdot \ln(G) + \mu_{V_{OC}} \cdot (T - T_{STC}) \tag{3}$$

$$R_s(G,T) = \frac{G_{STC} \cdot (V_{OC,STC} - V_{MPP,STC})}{4 \cdot (G \cdot (I_{MPP,STC} + \mu_{I_{SC}} \cdot (T - T_{STC})))} \tag{4}$$

$$R_{sh}(G,T) = \frac{2 \cdot G_{STC} \cdot (V_{MPP,STC} - \mu_{V_{OC}} \cdot (T - T_{STC}))}{(G \cdot (I_{SC,STC} - I_{MPP,STC}))} \tag{5}$$

As mentioned in the introduction, different mathematical models exist for calculating the output power of the PV module. One of those [10,29,33] is the empirical equation presented by (6), which will be further used to validate the double diode and experimental data.

$$P_{DC} = P_{STC} \cdot \left( \frac{G}{G_{STC}} \right) \cdot (1 + \mu_{P_{MPP}}(T - 25)) \tag{6}$$

where $\mu_{P_{MPP}}$ is the temperature coefficient of $P_{MPP}$ and $P_{STC}$ is the output power of PV module under STC conditions. All of the parameters that appear in (1)–(6) are presented in Table 2 for the considered mono-crystalline PV module for STC (PV Future—PVF 60M) [40].

**Table 2.** Electrical parameters of the considered mono-crystalline PV module for STC.

|  | PV Future—PVF 60M |
|---|---|
| Dimensions (l × w × s) [mm] | $993 \times 1648 \times 40$ |
| Cell size [mm$^2$] | $156 \times 156$ |
| $P_{MPP}$ [W] | 260 |
| $V_{MPP}$ [V] | 31 |
| $I_{MPP}$ [A] | 8.45 |
| $V_{OC}$ [V] | 37.8 |
| $I_{SC}$ [A] | 8.9 |
| $\mu_{I_{SC}}$ [%/°C] | 0.040 |
| $\mu_{V_{OC}}$ [%/°C] | $-0.330$ |
| $\mu_{P_{MPP}}$ [%/°C] | $-0.445$ |
| NOCT [°C] | 45 |
| Number of series connected cells | 60 |

### 2.3. Dynamic Temperature Model of the Photovoltaic Module

The dynamic thermal model of the PV module is presented differently within several papers [10,21,22,41–44]. The authors focus primarily on 1-D models of temperature distribution in PV module layers. Based on the thermal model of the PV module presented by [10], the dynamic heat balance presented in this paper consists of additional layers of Ethylene-vinyl acetate (EVA) foils. Therefore, the dynamic equations for all five layers, namely glass, EVA, PV cell, EVA and tedlar or PVF film-based backsheet, can be expressed by (7)–(11).

$$A_{PV} \cdot \rho_g \cdot d_g \cdot C_g \cdot \frac{dT_g}{dt} = A_{PV} \cdot (\alpha_g \cdot G - (1 - \phi_1) \cdot h_{conv.a-g} \cdot (T_g - T_a) - h_{cond.g-PV} \cdot (T_g - T_{PV}) \tag{7}$$

$$A_{\text{PV}} \cdot \rho_{\text{EVA}} \cdot d_{\text{EVA}} \cdot C_{\text{EVA}} \cdot \frac{dT_{\text{EVA}}}{dt} =$$
$$A_{\text{PV}} \cdot (\tau_{\text{g}} \cdot \alpha_{\text{EVA}} \cdot G - (h_{\text{cond.EVA}-\text{g}} \cdot (T_{\text{EVA}} - T_{\text{g}}) - h_{\text{cond.}EVA-\text{PV}} \cdot (T_{\text{EVA}} - T_{\text{PV}})) \tag{8}$$

$$A_{\text{PV}} \cdot \rho_{\text{PV}} \cdot d_{\text{PV}} \cdot C_{\text{PV}} \cdot \frac{dT_{\text{PV}}}{dt} =$$
$$A_{\text{PV}} \cdot (\tau_{\text{g}} \cdot \alpha_{\text{PV}} \cdot G \cdot FF - (h_{\text{cond.PV}-\text{g}} \cdot (T_{\text{PV}} - T_{\text{g}}) - h_{\text{cond.PV}-\text{t}} \cdot (T_{\text{PV}} - T_{\text{t}})) - P_{\text{DC}} \tag{9}$$

$$A_{\text{PV}} \cdot \rho_{\text{EVA}} \cdot d_{\text{EVA}} \cdot C_{\text{EVA}} \cdot \frac{dT_{\text{EVA}}}{dt} =$$
$$A_{\text{PV}} \cdot (\tau_{\text{g}} \cdot \alpha_{\text{EVA}} \cdot G - (h_{\text{cond.EVA}-\text{PV}} \cdot (T_{\text{EVA}} - T_{\text{PV}}) - h_{\text{cond.}EVA-\text{t}} \cdot (T_{\text{EVA}} - T_{\text{t}})) \tag{10}$$

$$A_{\text{PV}} \cdot \rho_{\text{t}} \cdot d_{\text{t}} \cdot C_{\text{t}} \cdot \frac{dT_{\text{t}}}{dt} =$$
$$A_{\text{PV}} \cdot (\tau_{\text{g}} \cdot \alpha_{\text{t}} \cdot G \cdot (1 - FF) - (1 - \phi_2) \cdot h_{\text{conv.t}-\text{a}} \cdot (T_{\text{t}} - T_{\text{a}}) - h_{\text{cond.PV}-\text{t}} \cdot (T_{\text{t}} - T_{\text{PV}}) \tag{11}$$

where $G$ is the solar radiation, $A_{\text{PV}}$ is the surface area of PV module, $\varphi_1$ is the heat flux ratio (equal to 0.46 [44]), $\varphi_2$ is the thermal radiation flux (equal to 0.52 [10]) between a tedlar layer and an ambient for the open-racked PV module, $T_{\text{a}}$ is the ambient temperature, $h_{\text{conv}}$ and $h_{\text{cond}}$ are the convective and conductive heat transfer coefficient (described in Appendix A), $P_{\text{DC}}$ is the output power, $FF$ is the fill factor and $\rho$, $d$, $C$, $T$, $\tau$ and $\alpha$ are the density, thickness, heat capacity, temperature, transmissivity and absorptivity of different layer, respectively. The indexes in the above-mentioned parameters describe the glass, EVA, PV cell and tedlar layers.

Thermal and mechanical parameters of the considered mono-crystalline PV module are presented in Table 3 (based on the literature [10,13,14,41]).

**Table 3.** Thermal and mechanical parameters of the considered mono-crystalline PV module.

|  | $\rho$ [kg/m$^3$] | $C$ [J/kgK] | $k$ [W/mK] | $d$ [mm] |
|---|---|---|---|---|
| Glass | 3000 | 500 | 1.8 | 4 |
| EVA | 960 | 2090 | 0.35 | 0.4 |
| PV cell | 2330 | 677 | 148 | 0.3 |
| Tedlar (PVF) | 1200 | 1250 | 0.2 | 0.4 |

## 3. Results

This section is divided into two subsections. The first subsection presents basic meteorological and experimental data: solar radiation $G$, wind speed $v$, ambient temperature $T_{\text{a}}$, direct current $I_{\text{DC}}$, direct voltage $V_{\text{DC}}$ and direct output power $P_{\text{DC}}$ of dual-axis PV tracking systems. The second section includes the result of thermal (temperature distribution) and electric (direct output power) models of the PV module.

*Meteorological and Experimental Data*

Validation of different models was performed based on meteorological and experimental data under outdoor conditions for 15 days in March. The three most common types of weather conditions were selected for comparison, namely: clear (2nd and 11th day), overcast (4th and 6th day) and cloudy (remaining 11 days). In fact, clear days are quite rare in Slovenia for the aforementioned month, so cloudy days are the most prevalent, which can be seen from the measurements of solar radiation shown in Figure 2. Furthermore, the wind speed in March is relatively low as well, with an average velocity of 1.55 m/s and a maximum velocity of 8.7 m/s.

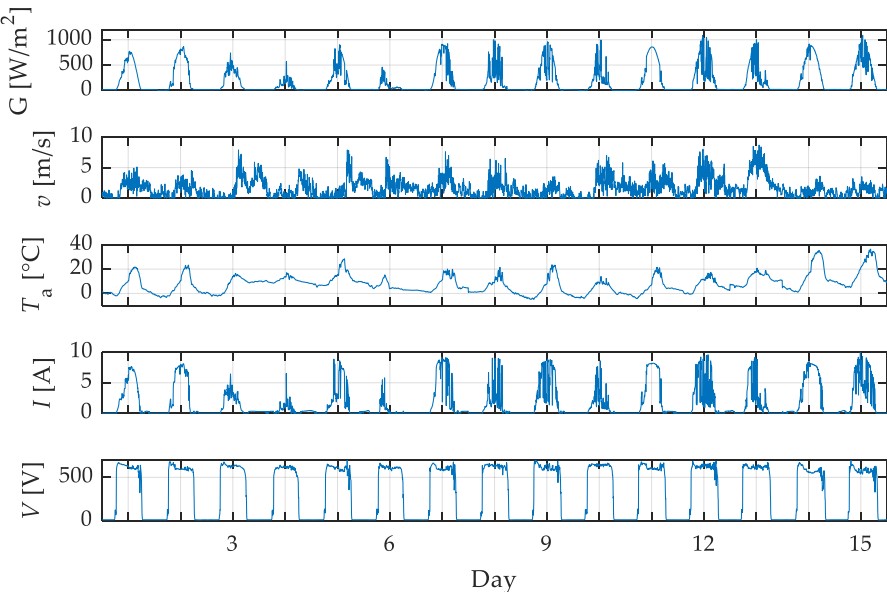

**Figure 2.** Meteorological and experimental data of the density of solar radiation $G$, wind speed $v$, ambient temperature $T_a$, current $I_{DC}$ and voltage $V_{DC}$.

Figure 3a presents the dynamic thermo-electric model created in Matlab/Simulink 2020b software by applying the s-function block diagram used for differential equations, while Figure 3b presents a static thermal model created in Ansys Transient Thermal 2020 R2 software for FEM analysis.

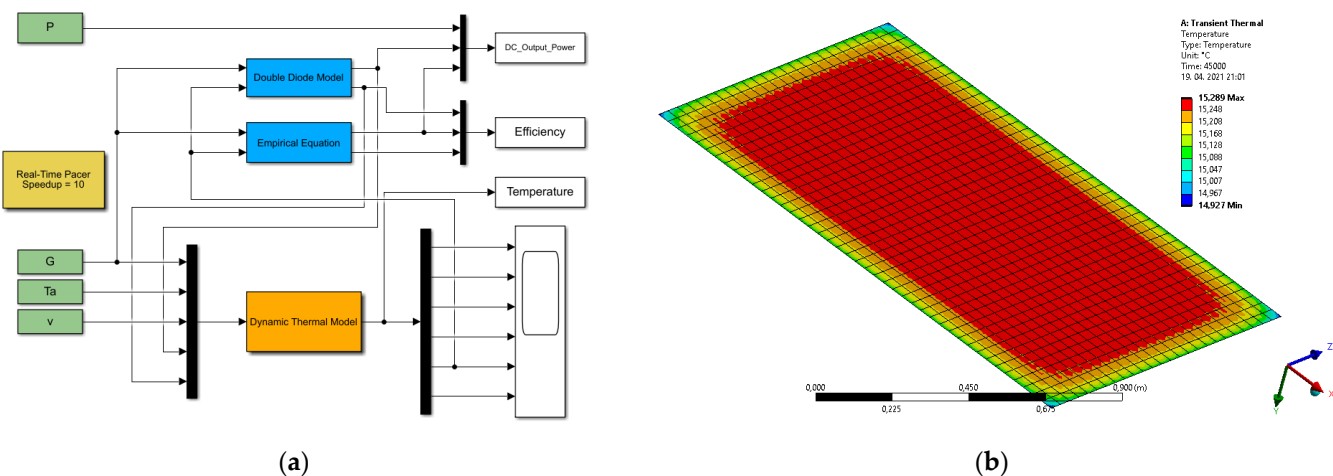

|  |  |
|:---:|:---:|
| (**a**) | (**b**) |

**Figure 3.** A thermo-electric model: (**a**) Dynamic—Matlab/Simulink; (**b**) Static—Ansys Transient Thermal.

The input parameters of the dynamic-thermal model made in Matlab/Simulink were solar radiation $G$, ambient temperature $T_a$, wind speed $v$, output power $P_{DC}$ and, fill factor $FF$, while the parameters for the dynamic electric model were solar radiation $G$ and operating temperature of the PV module $T$. On the other hand, the parameters of the static-thermal model made in Ansys Transient Thermal were solar radiation $G$, ambient temperature $T_a$ and, convective heat transfer coefficient $h_{conv}$. The validation of the existing dynamic model [10] with the static model was performed, as well. The results of both models are shown in Figures 4 and 5.

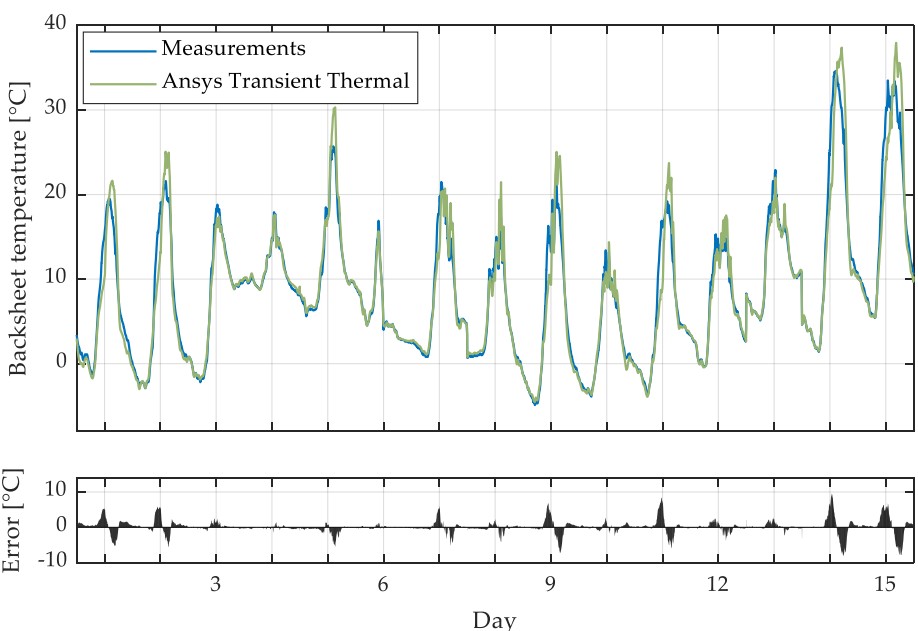

**Figure 4.** Validation of static thermal model (Ansys Transient Thermal) with measurements (for 15 days in March 2021).

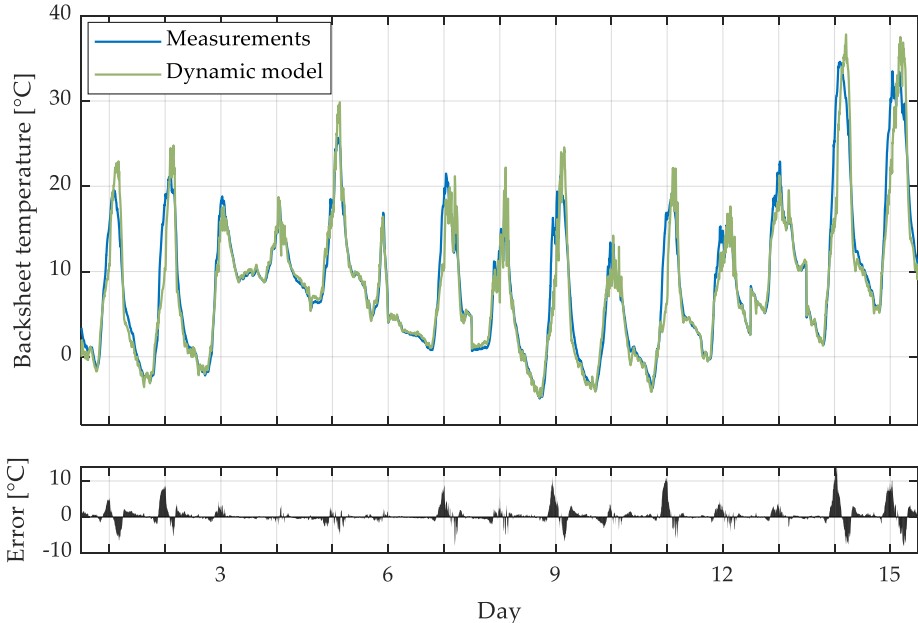

**Figure 5.** Validation of dynamic thermal model (Matlab/Simulink) with measurements (for 15 days in March 2021).

The largest temperature deviations occur more or less on days with a higher proportion of direct solar radiation (clear days). On average, the temperature error for 15 days is 0.32 °C (dynamic-thermal) and 0.63 °C (static-thermal). Figure 6 shows the RMSE and MAE values obtained for different weather conditions: clear, cloudy and overcast.

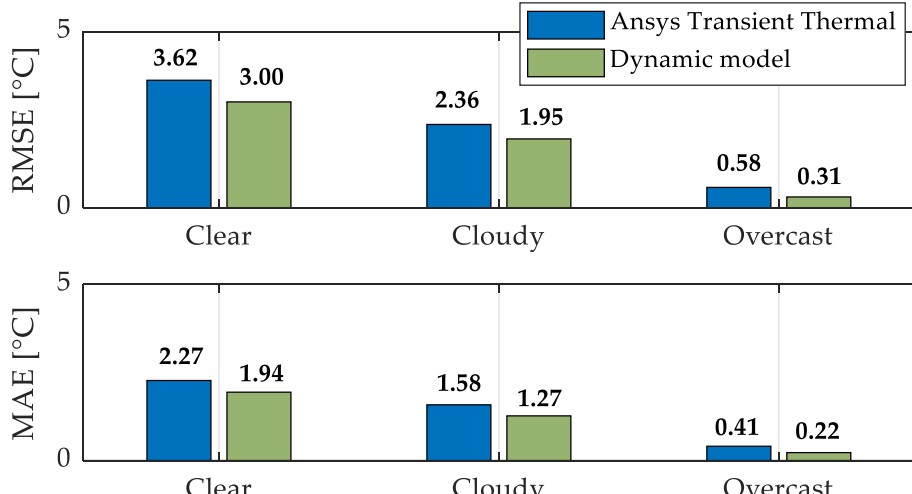

**Figure 6.** Comparison of static and dynamic thermal models for RMSE and MAE under different weather conditions.

It is clear that there are smaller temperature deviations in the case of the dynamic thermal model, as it is shown by the dynamic responses. Similar results are indicated by other authors [10,21,22]. It can be observed that on overcast days, there is a very small error of RMSE and MAE, due to smaller influence of direct solar radiation. In this case, the model is especially influenced by the ambient temperature $T_a$ and the wind speed $v$. Therefore, it was found that the pyranometers mounted on the dual-axis PV tracking systems measure the solar radiation with a significant error. The measurement error occurs due to non-compliance with the proposed calibration time by the manufacturer, which is also shown in Table 1 (calibration time). Temperature deviations are also affected by the missing diffuse component of solar radiation, which is not measured by pyranometers described in this study. Another important reason for the solar radiation measurement errors is dirt [45]. The dirt is accumulated on the pyranometer glass due to the weather conditions and must be cleaned once per year.

Influence of measurement error of solar radiation seen as an inner factor of the electric model for the calculation of the output power, will be presented below. However, as an essential part of the thermal models, optical loss (absorptivity, transmissivity and reflectivity), that significantly contributes to the increase or decrease of the proportion of solar radiation, strongly depends on the angle of incidence of the Sun's rays. In our case, the measurements were performed on a dual-axis PV tracking system that tracks the Sun's path (inclination and orientation) once per 4 min. Thus, a change in the inclination and orientation was predicted to calculate absorptivity, transmissivity and reflectivity by ±3° incident angle of the Sun's rays. Aforementioned, reflectivity, transmissivity and absorptivity are described by Snell's law as a function of the incident angle of the Sun's rays.

Meteorological and experimental data were also used for output power calculation and validation by using a dynamic electric model (double diode model) and already validated static electric model (empirical equation [10,29,33]). The validation results of both models are shown in Figures 7 and 8.

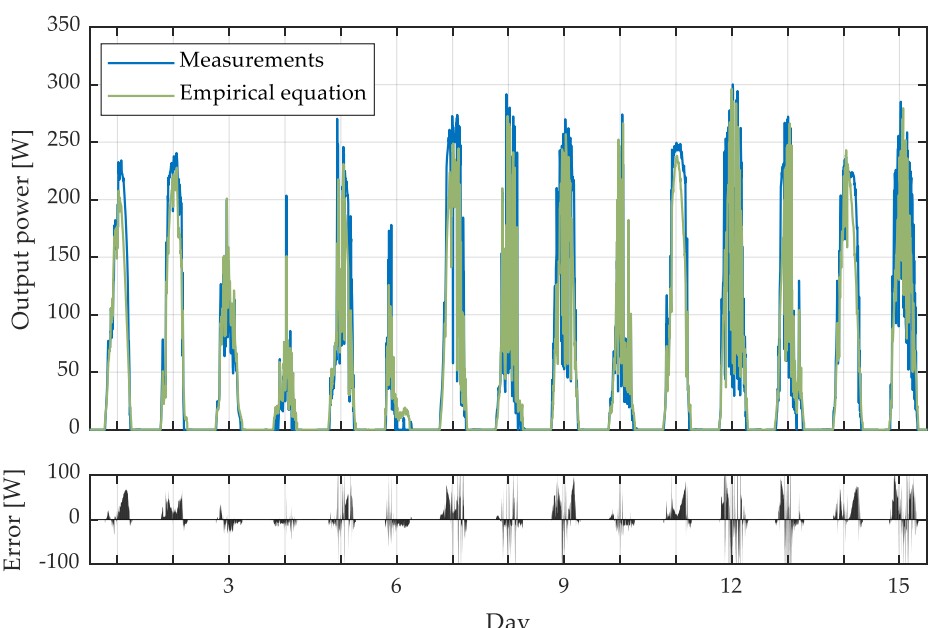

**Figure 7.** Validation of the static electric model (Matlab/Simulink-empirical equation) with experimental data (for 15 days in March 2021).

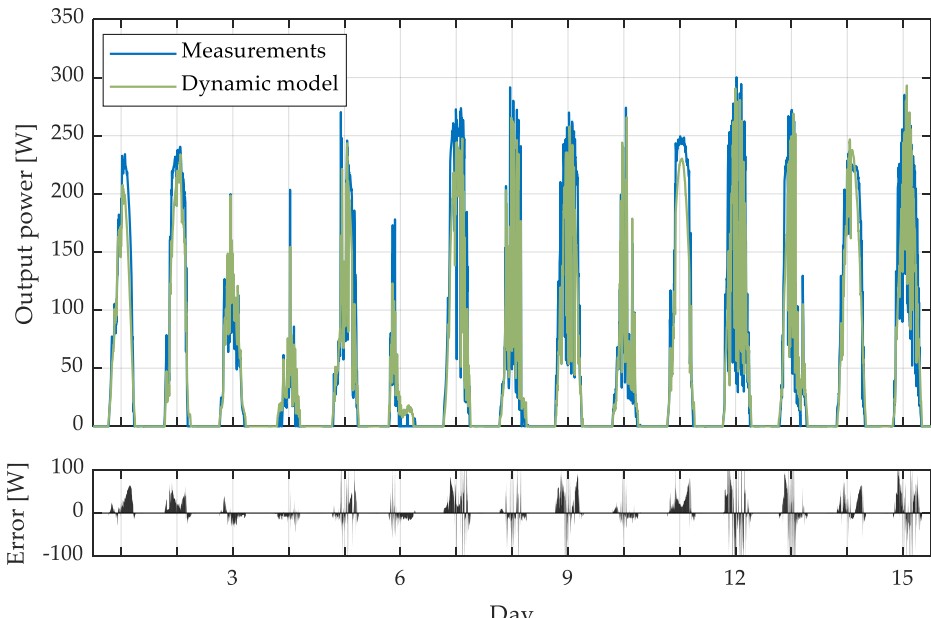

**Figure 8.** Validation of the dynamic electric model (Matlab/Simulink-double diode model) with experimental data (for 15 days in March 2021).

The largest output power deviations occur on days with a high to a medium proportion of direct solar radiation (clear and cloudy days). On average, the output power error for 15 days is 3.70 W (dynamic-electric model) and 4.06 W (static-electric model). Figure 9 shows the RMSE and MAE values obtained for different weather conditions: clear, cloudy and overcast.

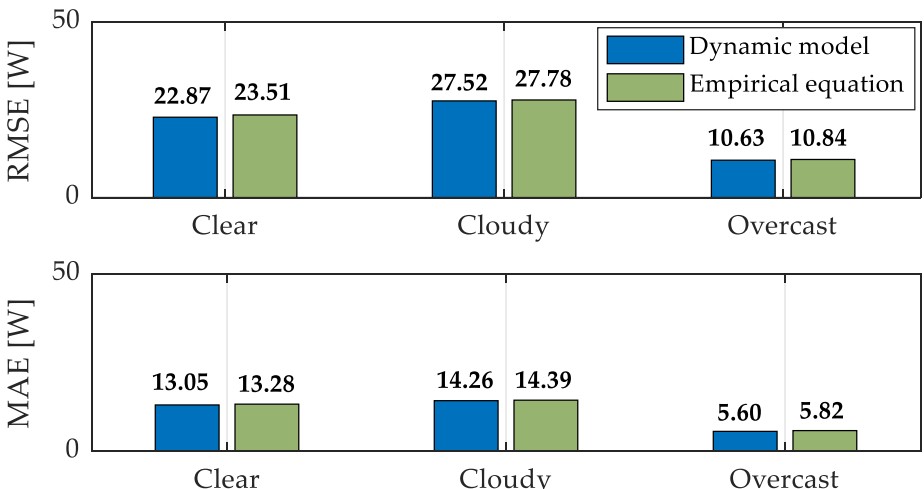

**Figure 9.** Comparison of the static and dynamic electric model for RMSE and MAE under different weather conditions.

For the output power calculation, the static model (empirical equation) represents a simple correlation between the output power $P$, solar radiation $G$ and operating temperature of the PV module $T$, while the double diode model considers a more accurate calculation, including several PV cell characteristics. However, the deviation between those two models is extremely small. The RMSE and MAE values differ by only 0.37 W and 0.19 W on average between the static and dynamic electric model. Solar radiation $G$ has the greatest influence on the output power of the PV module, followed by the operating temperature of the PV module $T$ and afterward, the air mass factor $AM$ (which was neglected due to the small influence). To explain the deviation of the output power, the actual measured temperature values of the PV module were used as input variables. It turned out that the output power was only 1%–2% more accurate, which confirms the accuracy of the thermal models. Based on the deviation between the calculated temperature distribution and the output power of the PV module, the assumption of incorrect measurements of solar radiation was confirmed. The output power of PV modules is also significantly affected by degradation, which reduces the output power of the PV module by 1% annually [46]. Degradation of PV modules involves many types of effects such as cell cracking, yellowing, browning and more [47,48]. The degradation rate takes effect when the output power of the PV module is reduced by at least 10%. However, high-efficiency PV modules with a 50% degradation rate can be more efficient than low-efficiency PV modules with a lower degradation rate [43]. Regardless of the error of measuring devices, the proposed dynamic thermo-electric model consisting of a dynamic thermal model presented in [10,21,22,41–44] and a dynamic electric model (double diode model) represent a good agreement between measured and simulated results.

## 4. Discussion

Our literature review has shown that there are currently quite a few different models (both static and dynamic) that summarize the temperature distribution in the PV module and/or the output power of the PV module. The models are presented dynamically in most cases, but very few models consist of a dynamic thermal and electrical part. Therefore, this paper aims to present a novel dynamic thermo-electric model of the PV module based on known equations involving different parameters given in various dependencies. In most studies, the temperature distribution in the PV module was presented for only one or three layers, while our study takes into account all five layers (glass, EVA, PV cell, EVA and tedlar). Optical losses of the thermal model are presented as a function of the incidence angle of the Sun's rays, which is very important for fixed PV modules. The RMSE and MAE values for cloudy and overcast days are in the same range or even better compared to the results of other studies [10,22,30], while more significant deviations are observed in the

case of sunny days due to measurement error. As described in Section 2.2, the calculation of the output power of the PV module is based on the double diode model. Some specific parameters are presented as a function of the solar radiation and the operating temperature of the PV module, which gives additional adaptability and accuracy to the model. The double diode model has already been very well presented in specific studies and represents an exact way of determining the output power of the PV module. From this point of view, the paper aimed to use the accuracy of a double diode model in combination with the dynamics of the thermal model to present a novel dynamic thermo-electric model of the PV module. Compared to the results of other studies [10], the double diode model results represent a more accurate assessment of the output power of the PV module, despite the measurement errors.

## 5. Conclusions

This paper presents a novel dynamic thermo-electric model of the PV module that consists of two interconnected parts via the operating temperature and output power of the PV module. The thermal model is described by heat balance equation for five different layers (glass, EVA, PV cell, EVA, tedlar), while the electric model is described by double diode model. The validation of the proposed model was made using the meteorological and experimental data of the presented dual-axis PV tracking system under outdoor conditions. Nevertheless, the thermal part of the model was additionally compared with the static model made in Ansys Transient Thermal, while the electric part was compared with a simple empirical equation for the output power calculation. The validation was made for three the most common types of weather conditions, namely clear, cloudy and overcast. The comparison of several models with each other was performed primarily due to the multi-criteria estimation and secondly due to the relatively large deviation between measured and simulated results. It was found that the measuring device (pyranometer) contains a measurement error due to several years of operation without calibration. However, the average RMSE and MAE values are 1.75 °C and 1.14 °C for the thermal part and 20.34 W and 10.97 W for the electrical part. The RMSE and MAE values differ between the static and dynamic thermal models by only 0.44 °C and 0.28 °C and between static and dynamic electric models by 0.37 W and 0.19 W on average. The advantage of the proposed model is that optical losses change as a function of the incident angle of the Sun's rays and are not a constant value, which is generally very important for a fixed PV system. Apart from that, it is important to note that most of the parameters (both thermal and electrical) are also presented as a function of the solar radiation *G* and the operating temperature of the PV module *T*. The presented dynamic thermo-electric model will be upgraded and used in future research areas, especially to determine the coolant temperature in PV/T modules.

**Author Contributions:** Conceptualization, K.S. and S.S.; methodology, K.S.; software, K.S.; validation, K.S.; formal analysis, K.S.; investigation, K.S.; resources, S.S., M.H. and B.Š.; data curation, K.S.; writing—original draft preparation, K.S.; writing—review and editing, S.S. and K.D.; visualization, K.S. and K.D.; supervision, S.S., M.H., K.D. and B.Š.; project administration, S.S., M.H. and B.Š.; funding acquisition, S.S., M.H. and B.Š. All authors have read and agreed to the published version of the manuscript.

**Funding:** This research and the APC was funded by Slovenian Research Agency under grants Applied Electromagnetics P2-0114.

**Institutional Review Board Statement:** Not applicable.

**Informed Consent Statement:** Not applicable.

**Conflicts of Interest:** The authors declare no conflict of interest. The funders had no role in the design of the study; in the collection, analyses or interpretation of data; in the writing of the manuscript or in the decision to publish the results.

## Nomenclature

| | |
|---|---|
| PV | photovoltaic |
| PV/T | photovoltaic/thermal |
| STC | standard test condition |
| RMSE | root mean square error |
| MAE | mean absolute error |
| NOCT | nominal operating cell temperature |
| MPPT | maximum power point tracking |
| EVA | ethylene-vinyl acetate |

Quantities used in equations:

| | |
|---|---|
| $A_{PV}$ | surface area of the PV module |
| $C$ | heat capacity |
| $d$ | thickness |
| $E_{g0}$ | bandwidth of cell material |
| $FF$ | fill factor |
| $G$ | solar radiation |
| $G_{STC}$ | input signal |
| $h_{cond}$ | conductive heat transfer coefficient |
| $h_{conv}$ | convective heat transfer coefficient |
| $I_{01}$ | reverse saturation current of the first diode |
| $I_{02}$ | reverse saturation current of the second diode |
| $I_{DC}$ | direct current |
| $I_{MPP,STC}$ | |
| $I_{ph}$ | photocurrent |
| $I_{SC}$ | short-circuit current |
| $I_{SC,STC}$ | maximal output current |
| $K$ | Boltzmann constant ($1.38065 \times 1023$ J/K) |
| $k$ | thermal conductivity |
| $K_i$ | extinction coefficient |
| $n$ | real refractive index |
| $n_1$ | the diode ideality factor of the first diode |
| $n_2$ | the diode ideality factor of the second diode |
| $N_S$ | number of series-connected PV cells |
| $P_{DC}$ | direct output power |
| $P_{STC}$ | output power under STC conditions |
| $q$ | electron charge ($1.602 \times 1019$ C) |
| $R_s$ | series resistance |
| $R_{sh}$ | shunt resistance |
| $T$ | operating temperature of the PV module |
| $T_a$ | ambient temperature |
| $T_{STC}$ | output signal |
| $v$ | wind speed |
| $V_{DC}$ | direct voltage |
| $V_{MPP,STC}$ | |
| $V_{OC}$ | open-circuit voltage |
| $V_{OC,STC}$ | open-circuit voltage under STC conditions |
| $V_{T1}$ | thermal voltage of the first diode |
| $V_{T2}$ | thermal voltage of the second diode |
| $\alpha$ | absorptivity |
| $\Theta_1$ | angle of incidence |
| $\Theta_2$ | angle of reflection |
| $\mu_{P_{MPP}}$ | temperature coefficient of $P_{MPP}$ |
| $\mu_{I_{SC}}$ | temperature coefficient of $I_{SC}$ |
| $\mu_{V_{OC}}$ | temperature coefficient of $V_{OC}$ |
| $\rho$ | density |
| $\tau_g$ | transmissivity |

| $\varphi_1$ | heat flux ratio |
| $\varphi_2$ | thermal radiation flux |

## Appendix A

This appendix is intended to provide an additional presentation of mathematical modeling described in Sections 2.2 and 2.3. A numerical method called Trapezoidal Rule —Backward Difference Formula of the 2nd order (TR-BDF2) was used to solve the implicit equation of the double diode model (described in Section 2.2). The photocurrent $I_{ph}$, the reverse saturation current of the first $I_{01}$ and second $I_{02}$ diode and the thermal voltage of the first $V_{T1}$ and second $V_{T2}$ diode is expressed by (A1)–(A6):

$$I_{ph} = \left( I_{SC} + \mu_{I_{SC}}(T - T_{STC}) \right) \cdot \frac{G}{G_{STC}} \tag{A1}$$

$$I_{ph} = I_{SC} \tag{A2}$$

$$I_{01} = \left( \frac{I_{SC}}{\left( \exp^{\frac{V_{OC} \cdot q}{T \cdot K \cdot N_S \cdot n_1}} - 1 \right)} \right) \cdot \left( \frac{T}{T_{STC}} \right)^3 \cdot \exp^{\left( \frac{q \cdot E_{g0} \cdot \left( \frac{1}{T_{STC}} - \frac{1}{T} \right)}{n_1 \cdot K} \right)} \tag{A3}$$

$$I_{02} = \left( \frac{I_{SC}}{\left( \exp^{\frac{V_{OC} \cdot q}{T \cdot K \cdot N_S \cdot n_2}} - 1 \right)} \right) \cdot \left( \frac{T}{T_{STC}} \right)^3 \cdot \exp^{\left( \frac{q \cdot E_{g0} \cdot \left( \frac{1}{T_{STC}} - \frac{1}{T} \right)}{n_2 \cdot K} \right)} \tag{A4}$$

$$V_{T1} = \frac{n_1 \cdot K \cdot T}{q} \tag{A5}$$

$$V_{T2} = \frac{n_2 \cdot K \cdot T}{q} \tag{A6}$$

where $E_{g0}$ is the bandwidth of cell material (1.21 eV), while $n_1$ and $n_2$ are the ideality factors of the first and second diode. The ideality factors $n_1$ and $n_2$ represent the diffusion and recombination current components for the double diode model, while their values range between 1 (ideal diode) and 2 (more realistic diode). In this study, the values of $n_1$ and $n_2$ were 1 and 1.2, respectively.

The convective $h_{conv}$ and conductive $h_{cond}$ heat transfer coefficients are expressed by (A7)–(A8) [49,50]:

$$h_{conv} = 5.7 - 3.8 \cdot v \tag{A7}$$

$$h_{cond.x-y} = \left( \frac{d_x}{k_x} + \frac{d_y}{k_y} \right)^{-1} \tag{A8}$$

where $d$ is the layer thickness, $k$ is the thermal conductivity of layers. The indexes $x$ and $y$ describes the current and next layer of the PV module. The absorptivity of the glass layer $\alpha_g$ is expressed by (A9) [51]:

$$\alpha_g = 1 - \tau_a \tag{A9}$$

The transmissivity of the glass layer $\tau_a$ is expressed by (A10):

$$\tau_a = \exp^{-\frac{K_i \cdot d}{\cos \Theta_2}} \tag{A10}$$

where $K_i$ is extinction coefficient ($K_{i,glass} = 3.55 \times 10^{-6}$; $K_{i,EVA} = 2.34 \times 10^{-6}$), $d$ is the thickness of a layer and $\Theta_2$ is the angle of reflection (expressed by (A11)) [52].

$$\Theta_2 = \sin^{-1} \left( \frac{\sin \Theta_1}{n} \right) \tag{A11}$$

where $\Theta_1$ is the angle of incidence and $n$ is the real refractive index ($n_{glass} = 1.52$; $n_{EVA} = 1.48$). $\tau$ is the transmissivity that considers multiple refractions and reflections between glass layer and PV cell, and is expressed by (A13):

$$\tau_r = \frac{1}{2}\left( \frac{1 - \frac{\tan(\Theta_2 - \Theta_1)}{\tan(\Theta_2 + \Theta_1)}}{1 + \frac{\tan(\Theta_2 - \Theta_1)}{\tan(\Theta_2 + \Theta_1)}} + \frac{1 - \frac{\sin(\Theta_2 - \Theta_1)}{\sin(\Theta_2 + \Theta_1)}}{1 + \frac{\sin(\Theta_2 - \Theta_1)}{\sin(\Theta_2 + \Theta_1)}} \right) \tag{A12}$$

$$\tau = \tau_a \cdot \tau_r \tag{A13}$$

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
