# Peer review of "Experimental Validation of a Thermo-Electric Model of the Photovoltaic Module under Outdoor Conditions"

_applsci, doi:10.3390/app11115287_

Round 1

Reviewer 1 Report

The topic of this paper is interesting to the readers, within the scope of the journal, however prior its publication changes must be done.

The use of English must be improved. The paper includes several grammatical and syntax errors.

The Introduction must be revised. The authors must present the general research area to unfamiliar readers and at most to present the current state-of-the-art in order to show the contribution/novelty of their work. Authors must describe/analyse more the current mentioned references and must include many more related references, such as the following:

Seritan G.-C., Enache B.-A., Adochiei F.-C., Argatu F.C., Christodoulou C., Vita V., Toma A.R., Gandescu C.H., Hathazi F.-I., Performance evaluation of photovoltaic panels containing cells with different bus bars configurations in partial shading conditions, Revue Roumaine des Sciences Techniques - Électrotechnique et Énergétique, Vol. 65, No. 1-2, 2020, pp. 67-70.

Damianaki K., Christodoulou C., Kokalis C.-C.A., Kyritsis A., Ellinas E.D., Vita V., Gonos I.F., Lightning protection of photovoltaic systems: Computation of the developed potentials, Applied Sciences, Vol. 11, No. 1, (DOI) 10.3390/app11010337, 2021.

A separate discussion section that will comment on the produced results must be included.

Conclusions must summarize the work presented within the paper.

Author Response

Summary of corrections made in the paper

Experimental validation of a thermo-electric model of the photovoltaic module under outdoor conditions

(Manuscript ID: applsci-1218204)

We thank the reviewers for their valuable comments on our paper, which have substantially improved the quality of our paper. We have made a concerted effort to address all the reviewers concerns. Below we quote and respond to each reviewer’s comments.

Reviewer 1:

The topic of this paper is interesting to the readers, within the scope of the journal, however prior its publication changes must be done.

  1. The use of English must be improved. The paper includes several grammatical and syntax errors.

Answer 1:

Thank you. Corrected. The paper was proofread by a native-english speaker.

  1. The Introduction must be revised. The authors must present the general research area to unfamiliar readers and at most to present the current state-of-the-art in order to show the contribution/novelty of their work. Authors must describe/analyse more the current mentioned references and must include many more related references, such as the following:

Seritan G.-C., Enache B.-A., Adochiei F.-C., Argatu F.C., Christodoulou C., Vita V., Toma A.R., Gandescu C.H., Hathazi F.-I., Performance evaluation of photovoltaic panels containing cells with different bus bars configurations in partial shading conditions, Revue Roumaine des Sciences Techniques - Électrotechnique et Énergétique, Vol. 65, No. 1-2, 2020, pp. 67-70.

Damianaki K., Christodoulou C., Kokalis C.-C.A., Kyritsis A., Ellinas E.D., Vita V., Gonos I.F., Lightning protection of photovoltaic systems: Computation of the developed potentials, Applied Sciences, Vol. 11, No. 1, (DOI) 10.3390/app11010337, 2021.

Answer 2:

Thank you. Corrected. The introduction has been supplemented with additional content that will make it easier for readers to read and review. Due to the relatively long introduction, I divided this section into three parts, namely: main introduction, »Literature review of the existing studies« and »Aims and specifics of the current research«. Additional references has also been taken into account.

  1. A separate discussion section that will comment on the produced results must be included.

Answer 3:

Thank you. Corrected.

The discussion section has been added (Section 4). In this section, were presented and discussed the obtained results, which were later compared with the results of other similar studies.

  1. Conclusions must summarize the work presented within the paper.

Answer 4:

Thank you for your comment.

The section conclusion summarizes the work presented in the paper. The discussion was urgently needed, but I do not see the necessary changes in this part.

Reviewer 2 Report

The model equations are known. Most of the model parameters are not provided which is very important as they impact the results. The experimental data are not very significant to show the temperature influence on power due to the chosen period and to sensor deviation. I don't see any relevant added value for the readers.

Author Response

Summary of corrections made in the paper

Experimental validation of a thermo-electric model of the photovoltaic module under outdoor conditions

(Manuscript ID: applsci-1218204)

We thank the reviewers for their valuable comments on our paper, which have substantially improved the quality of our paper. We have made a concerted effort to address all the reviewers concerns. Below we quote and respond to each reviewer’s comments.

Reviewer 2:

  1. The model equations are known. Most of the model parameters are not provided which is very important as they impact the results. The experimental data are not very significant to show the temperature influence on power due to the chosen period and to sensor deviation. I don't see any relevant added value for the readers.

Answer 1:

Thank you. Corrected.

The values of several parameters were added in the paper (especially in Appendix I).

An essential part of this paper is presenting the validation results of the presented thermos-electric model with measurements. Therefore less attention was paid to the influence of the operating temperature of the PV module on the output power.

Reviewer 3 Report

Minor Revision

In this manuscript, the author presented an interested paper which presents a dynamic thermo-electric model of the PV module which consists of coupled thermal and electric parts. Whereas the thermal part is described by heat balance equation for five layers, while the electric part is described by a double-diode model. The author validate this model by implement using the PV tracking system under outdoor conditions, and compare the results with the static model made in Ansys Transient Thermal and empirical equation for the output power calculation under 3 typical weather conditions.

In general, this is a good manuscript. However, there are some issues need to be addressed/clarified.

  1. Line 120, the author needs to clarify the use of a Power Meter and Hall effect sensor. Is it for DC and AC measurement separately?
  2. Line 121, might be “a several-local-and-one-global PLC”.
  3. Line 123, should be “are sampled every 5 minutes”.
  4. Line 155, the author need to add reference for “One of those”.
  5. Line 166, should be “by [3], ”
  6. Line 193, the author need to describe it clearly how Figure 2 indicate “clear days are quite rare in Slovenia for aforementioned month, so cloudy days are the most prevalent”.
  7. Figure 3, the font size needs to be larger. Currently it is hard to see.
  8. Line 211, what does the author mean “appropriate result”?
  9. Line 234, need to clarify why “is also evident from Table 1 (calibration time)”.
  10. Line 236, grammar mistake. It can be “Another important reason xxx is dirt.”
  11. Line 240, not clear what the author wants to say. Can be something like “will be seen as an inner factor of the electric model?”
  12. Line 241, grammar mistake. Can be “as an essential part xxx, the optical loss that significantly contributes to xxx, strongly depends on xxx”.
  13. The author mentioned “The advantage of the proposed model is that the optical losses change as a function of the incident angle of the Sun’s rays and not as a constant value, which is generally very important for a fixed PV system. However, the manuscript does not show the importance of this advantage. How does the proposed model makes a difference including this consideration. It would be helpful to show the strength of the proposed by comparing the results predicted of the proposed model and other predicted models.

Author Response

Summary of corrections made in the paper

Experimental validation of a thermo-electric model of the photovoltaic module under outdoor conditions

(Manuscript ID: applsci-1218204)

We thank the reviewers for their valuable comments on our paper, which have substantially improved the quality of our paper. We have made a concerted effort to address all the reviewers concerns. Below we quote and respond to each reviewer’s comments.

Reviewer 3:

In this manuscript, the author presented an interested paper which presents a dynamic thermo-electric model of the PV module which consists of coupled thermal and electric parts. Whereas the thermal part is described by heat balance equation for five layers, while the electric part is described by a double-diode model. The author validate this model by implement using the PV tracking system under outdoor conditions, and compare the results with the static model made in Ansys Transient Thermal and empirical equation for the output power calculation under 3 typical weather conditions.

In general, this is a good manuscript. However, there are some issues need to be addressed/clarified.

  1. Line 120, the author needs to clarify the use of a Power Meter and Hall effect sensor. Is it for DC and AC measurement separately?

Answer 1:

Thank you. Corrected.

Yes, Power Meter is for AC measurements, while Hall effect sensor is for DC measurements.

The DC/AC inverter does not include measurements of AC and DC current, voltage and output power data, therefore a Power Meter (Siemens SENTROM PAC4200 – AC measurements) and a Hall Effect Sensor (T201DCH100 – DC measurements) are additionally installed.

  1. Line 121, might be “a several-local-and-one-global PLC”.

Answer 2:

Thank you. Corrected.

The entire system is connected to a several-local-and-one-global PLC, while the data is displayed in a SCADA system, that allows an easy monitoring and management.

  1. Line 123, should be “are sampled every 5 minutes”.

Answer 3:

Thank you. Corrected.

All of the meteorological and experimental data are sampled every 5 minutes.

  1. Line 155, the author need to add reference for “One of those”.

Answer 4:

Thank you. Corrected.

One of those [3,21,24] is the empirical equation presented by (6), which will be further used to validate the double-diode and experimental data.

  1. Line 166, should be “by [3], ”

Answer 5:

Thank you. Corrected.

Based on the thermal model of the PV module presented by [3], the dynamic heat balance presented in this paper consists of an additional layers of Ethylene-vinyl acetate (EVA) foils.

  1. Line 193, the author need to describe it clearly how Figure 2 indicate “clear days are quite rare in Slovenia for aforementioned month, so cloudy days are the most prevalent”.

Answer 6:

Thank you. Corrected.

In fact, clear days are quite rare in Slovenia for the aforementioned month, so cloudy days are the most prevalent, which can be seen from the measurements of solar radiation shown in Figure 2.

  1. Figure 3, the font size needs to be larger. Currently it is hard to see.

Answer 7:

Thank you. Corrected.

  1. Line 211, what does the author mean “appropriate result”?

Answer 8:

Thank you. Corrected. This adjective was nothing special, it was obviously a grammatical error.
The results of both models are shown in Figures 4 and 5.

  1. Line 234, need to clarify why “is also evident from Table 1 (calibration time)”.

Answer 9:

Thank you. Corrected.

Therefore, it was found that the pyranometers mounted on the dual-axis PV tracking sys-tems measure the solar radiation with a significant error. The measurement error occurs due to non-compliance with the proposed calibration time by the manufacturer, which is also shown in Table 1 (calibration time).

  1. Line 236, grammar mistake. It can be “Another important reason xxx is dirt.”

Answer 10:

Thank you. Corrected.
Another important reason for the solar radiation measurement errors is dirt [36]. The dirt is accumulated on the pyranometer glass due to the weather conditions and must be cleaned once per year.

  1. Line 240, not clear what the author wants to say. Can be something like “will be seen as an inner factor of the electric model?”

Answer 11:

Thank you. Corrected.
In the following, the influence of measurement error of solar radiation will be seen as an inner factor of the electric model for the calculation of the output power.

  1. Line 241, grammar mistake. Can be “as an essential part xxx, the optical loss that significantly contributes to xxx, strongly depends on xxx”.

Answer 12:

Thank you. Corrected.

Influence of measurement error of solar radiation seen as an inner factor of the electrical model for the calculation of the output power, will be presented below. However, as an essential part of the thermal models, optical loss (absorptivity, transmissivity and reflectivity), that significantly contributes to the increase or decrease of the proportion of solar radiation, strongly depends on the angle of incidence of the Sun's rays.

  1. The author mentioned “The advantage of the proposed model is that the optical losses change as a function of the incident angle of the Sun’s rays and not as a constant value, which is generally very important for a fixed PV system. However, the manuscript does not show the importance of this advantage. How does the proposed model makes a difference including this consideration. It would be helpful to show the strength of the proposed by comparing the results predicted of the proposed model and other predicted models.

Answer 13:

Thank you for your comment.

This is a well-known fact that the use of parameters in dependence gives more accurate or more realistic values than in comparison with constant values. In this study, the comparison between other models would be somewhat demanding in terms of importance, as the results or measurements are given for dual-axis PV tracking system, which always has a constant angle of incidence of sunlight and consequently other optical loss values. The comparison would be meaningful and interesting if other studies used a similar model on a PV tracking system.

Reviewer 4 Report

  • Regarding the format of the document, some suggestions are as follows.

Keywords must be separated by semicolons.

English must be revised due to the fact that there are some typos concerning plural and singular. For example, in line 154, “… a different mathematical models exists for …” is not properly written.

In line 201 “ANSYS” is found, whereas “Ansys” is used in the rest of the manuscript.

In figure 3 a) the term “emperical” appears, which seems to be a mistake.

The format of references must be revised following the template. For example, the abbreviated names of journals must be used.

  • About the content of the manuscript, these issues are commented.

In the keywords, terms like “PV module” or “photovoltaic” could be included.

The introductory section consists on a single and excessively long paragraph. It should be divided into some paragraph to facilitate reading.

As the authors comment, the real temperature of PV cells is higher than that measured with temperature probes or sensors. In fact, there is some research in this regard. For instance, the work of King et al. is commonly cited to reflect such a difference:

  • King, D., Boyson, W., Kratochvil, J.: Photovoltaic array performance model. Tech. rep., SAND2004-3535, /http://www.sandia.gov/pv/docs/PDF/King%20SANDS.pdf

Regarding the estimation of PV cells from weather conditions like ambient temperature and incident irradiance, some well-known works have been left unnoticed. Namely, the following models constitute essential references in the topic:

  • Ross AG. Flat-plate photovoltaic module and array engineering. In: Proc. 1982, Annu. Meet. Am. Sect. Int. Sol. Energy Soc., Houston, Texas; 1982. p. 4321–4.
  • Faiman D. Assessing the outdoor operating temperature of photovoltaic modules. Prog Photovoltaics Res Appl 2008;16:307–15. https://doi.org/10.1002/pip.813.

Moreover, aiming at enhancing the contextualization, the increasing application and investigation of PV generation in smart grids and microgrids could be slightly mentioned, if the authors agree. Some papers recently published in the MDPI dealing with these topics are suggested:

  • Garcia-Torres, F.; Vazquez, S.; Moreno-Garcia, I.M.; Gil-de-Castro, A.; Roncero-Sanchez, P.; Moreno-Munoz, A. Microgrids Power Quality Enhancement Using Model Predictive Control. Electronics 2021, 10, 328. https://doi.org/10.3390/electronics10030328.
  • González, I.; Calderón, A.J.; Portalo, J.M. Innovative Multi-Layered Architecture for Heterogeneous Automation and Monitoring Systems: Application Case of a Photovoltaic Smart Microgrid. Sustainability 2021, 13, 2234. https://doi.org/10.3390/su13042234.
  • Dairi, A.; Harrou, F.; Sun, Y.; Khadraoui, S. Short-Term Forecasting of Photovoltaic Solar Power Production Using Variational Auto-Encoder Driven Deep Learning Approach. Appl. Sci. 2020, 10, 8400. https://doi.org/10.3390/app10238400.
  • Wang, J.; Li, K.-J.; Liang, Y.; Javid, Z. Optimization of Multi-Energy Microgrid Operation in the Presence of PV, Heterogeneous Energy Storage and Integrated Demand Response. Appl. Sci. 2021, 11, 1005. https://doi.org/10.3390/app11031005.

In subsection 2.1 the experimental setup is described, where automation-related equipment is scarcely commented. Namely, a PLC and a supervisory system are mentioned, but they are underestimated given their relevant role in the operation and surveillance of the PV tracker. Therefore, it would be desirable to mention, at least, the models of hardware and software involved in such equipment for a proper description.

Concerning the Matlab/Simulink environment, the specific version and toolboxes that have been used should be given for the interested reader.

As the authors know, the ideality factor, n, is one of the parameters of the equivalent circuit model, both using one or two diodes. In this sense, it is quite surprising that there is no comment about the value that this parameter adopts. Even, there are two ideality factors due to the fact that the two-diode model is managed. To solve this issue, the authors should explain that this value is between 1 (ideal diode) and 2 (more realistic diode), as well as indicating the values that they have considered to implement their model.

Equation 12 assumes that the photo-generated current is equivalent to the short circuit current. This is a common approximation, but the authors should explicitly indicate such assumption for a clearer presentation of the model. This equation belongs to the Appendix I.

Table 2 reports the electrical features and parameters of the studied PV module. It is advised to mention that these values are provided by the manufacturer for STC.

The electrical model is scarcely presented in subsection 2.2 but there is no explanation about how the parameters of such a model are adjusted. The reported model is implicit, so it requires solving techniques like numerical methods. If empirical IV curves have been used to adjust some of the parameters, for example the parasitic resistances, it should be clearly expounded.

In lines 281-282 the effect of degradation in power generation is slightly commented. In this regard, that statement could be enriched by mentioning some effects of degradation like cell cracking, yellowing, browning, etc.

The achieved results are validated on the view of error metrics (RMSE and MAE); however, there is no comparison with previous literature. Moreover, the common practice when modeling PV cells/modules under the electric equivalent circuit is obtaining the estimation of generated current, so the RMSE is obtained for such magnitude.

 In addition, it is unclear the novelty or contribution of the reported research. The authors should emphasize the novelties and benefits in order to state the relevance and opportunity of their research. This must be mainly highlighted in the discussion or conclusions.

Author Response

Summary of corrections made in the paper

Experimental validation of a thermo-electric model of the photovoltaic module under outdoor conditions

(Manuscript ID: applsci-1218204)

We thank the reviewers for their valuable comments on our paper, which have substantially improved the quality of our paper. We have made a concerted effort to address all the reviewers concerns. Below we quote and respond to each reviewer’s comments.

Reviewer 4:

Regarding the format of the document, some suggestions are as follows.

  1. Keywords must be separated by semicolons.

Answer 1:

Thank you, but the keywords have already been separated by semicolons.

  1. English must be revised due to the fact that there are some typos concerning plural and singular. For example, in line 154, “… a different mathematical models exists for …” is not properly written.

Answer 2:

Thank you. The paper was proofread by a native-english speaker.

As mentioned in the introduction, different mathematical models exist for calculating the output power of the PV module.

  1. In line 201 “ANSYS” is found, whereas “Ansys” is used in the rest of the manuscript.

Answer 3:

Thank you. Corrected.

Figure 3a presents the dynamic thermo-electric model created in Matlab/Simulink software applying the s-function block diagram used for differential equations, while fig-ure 3b presents a static thermal model created in Ansys Transient Thermal software for FEM analysis.

  1. In figure 3 a) the term “emperical” appears, which seems to be a mistake.

Answer 4:

Thank you. Corrected.

  1. The format of references must be revised following the template. For example, the abbreviated names of journals must be used.

Answer 5:

Thank you. Corrected.

  1. About the content of the manuscript, these issues are commented.

In the keywords, terms like “PV module” or “photovoltaic” could be included.

Answer 6:

Thank you. Corrected.

Keywords: dynamic modeling; thermo-electric model; accuracy; measuring device; temperature, output power; PV module

  1. The introductory section consists on a single and excessively long paragraph. It should be divided into some paragraph to facilitate reading.

Answer 7:

Thank you. Corrected.

The introduction was divided into three subsections, namely: main introduction, »Literature review of the existing studies« and »Aims and specifics of the current research«. In addition, the introduction was further improved with additional references.

  1. As the authors comment, the real temperature of PV cells is higher than that measured with temperature probes or sensors. In fact, there is some research in this regard. For instance, the work of King et al. is commonly cited to reflect such a difference:

King, D., Boyson, W., Kratochvil, J.: Photovoltaic array performance model. Tech. rep., SAND2004-3535, /http://www.sandia.gov/pv/docs/PDF/King%20SANDS.pdf

Answer 8:

Thank you. Added.

The proposed reference was added under the introduction subsection »Literature review of the existing studies«, where we discussed the results of the study.

Furthermore, Sandia National Laboratories [20] proposed a new empirically-based thermal model for flat-plate PV modules mounted in an open rack. The thermal model consists of temperature calculation of the backsheet layer and PV cell. This model has proven to be adaptable and adequate for flat-plate PV modules with an accuracy of ±5 °C. The results show that the temperature of the PV cell and the temperature of the backsheet layer can differ significantly, especially for different types of PV modules.

  1. Regarding the estimation of PV cells from weather conditions like ambient temperature and incident irradiance, some well-known works have been left unnoticed. Namely, the following models constitute essential references in the topic:

Ross AG. Flat-plate photovoltaic module and array engineering. In: Proc. 1982, Annu. Meet. Am. Sect. Int. Sol. Energy Soc., Houston, Texas; 1982. p. 4321–4.

Faiman D. Assessing the outdoor operating temperature of photovoltaic modules. Prog Photovoltaics Res Appl 2008;16:307–15. https://doi.org/10.1002/pip.813.

Answer 9:

Thank you. Added.

The proposed references were added under the introduction subsection »Literature review of the existing studies«, where we discussed the results of the studies.

Since temperature distribution in the PV module is described differently for many types and designs, Ross [18] provides an overview of design requirements, design analysis, and test methods for flat-plate PV modules. At that time, the NOCT model was usually used for temperature assessment of the PV module. Similarly, Faiman [19] presents the flat-plate PV module outdoor operating temperature assessment using a simple modified form of the Hottel-Whillier-Bliss (HWB) equation.

  1. Moreover, aiming at enhancing the contextualization, the increasing application and investigation of PV generation in smart grids and microgrids could be slightly mentioned, if the authors agree. Some papers recently published in the MDPI dealing with these topics are suggested:

Garcia-Torres, F.; Vazquez, S.; Moreno-Garcia, I.M.; Gil-de-Castro, A.; Roncero-Sanchez, P.; Moreno-Munoz, A. Microgrids Power Quality Enhancement Using Model Predictive Control. Electronics 2021, 10, 328. https://doi.org/10.3390/electronics10030328.

González, I.; Calderón, A.J.; Portalo, J.M. Innovative Multi-Layered Architecture for Heterogeneous Automation and Monitoring Systems: Application Case of a Photovoltaic Smart Microgrid. Sustainability 2021, 13, 2234. https://doi.org/10.3390/su13042234.

Dairi, A.; Harrou, F.; Sun, Y.; Khadraoui, S. Short-Term Forecasting of Photovoltaic Solar Power Production Using Variational Auto-Encoder Driven Deep Learning Approach. Appl. Sci. 2020, 10, 8400. https://doi.org/10.3390/app10238400.

Wang, J.; Li, K.-J.; Liang, Y.; Javid, Z. Optimization of Multi-Energy Microgrid Operation in the Presence of PV, Heterogeneous Energy Storage and Integrated Demand Response. Appl. Sci. 2021, 11, 1005. https://doi.org/10.3390/app11031005.

Answer 10:

Thank you. Added.

The proposed references were added as part of the main introduction.

Nowadays, due to self-sufficiency, the off-grid system is a very common way of connection, as it presents a great energy potential in underdeveloped countries or in countries without a well-established electricity network. To facilitate the digital transformation of energy infrastructure, it is also necessary to encourage the development and introduction of smart grids [2-4] and microgrids [5].

  1. In subsection 2.1 the experimental setup is described, where automation-related equipment is scarcely commented. Namely, a PLC and a supervisory system are mentioned, but they are underestimated given their relevant role in the operation and surveillance of the PV tracker. Therefore, it would be desirable to mention, at least, the models of hardware and software involved in such equipment for a proper description.

Answer 11:

Thank you. Corrected.

Unfortunately, we were unable to obtain the information about the programming software. The SCADA system, which is already mentioned in the description, is available to users for capturing and displaying data.

The entire system is connected to a several-local-and-one-global PLC (S7-300), while the data is displayed in a SCADA software system that allows easy monitoring and man-agement.

  1. Concerning the Matlab/Simulink environment, the specific version and toolboxes that have been used should be given for the interested reader.

Answer 12:

Thank you. Corrected.

No additional toolboxes were used for this simulation model.

Figure 3a presents the dynamic thermo-electric model created in Matlab/Simulink 2020b software by applying the s-function block diagram used for differential equations, while figure 3b presents a static thermal model created in Ansys Transient Thermal 2020 R2 software for FEM analysis.

  1. As the authors know, the ideality factor, n, is one of the parameters of the equivalent circuit model, both using one or two diodes. In this sense, it is quite surprising that there is no comment about the value that this parameter adopts. Even, there are two ideality factors due to the fact that the two-diode model is managed. To solve this issue, the authors should explain that this value is between 1 (ideal diode) and 2 (more realistic diode), as well as indicating the values that they have considered to implement their model.

Answer 13:

Thank you. Corrected.

An additional explanation was given in Appendix I.

The ideality factors n1 and n2 represent the diffusion and recombination current components for the double diode model, while their values range between 1 (ideal diode) and 2 (more realistic diode). In this study, the values of n1 and n2 were 1 and 1.2, respectively.

  1. Equation 12 assumes that the photo-generated current is equivalent to the short circuit current. This is a common approximation, but the authors should explicitly indicate such assumption for a clearer presentation of the model. This equation belongs to the Appendix I.

Answer 14:

Thank you. Corrected.

The approximation between the photo-generated current and the short-circuit current was added to the Appendix I as equation 13.

  1. Table 2 reports the electrical features and parameters of the studied PV module. It is advised to mention that these values are provided by the manufacturer for STC.

Answer 15:

Thank you. Corrected.

All of the parameters that appear in (1)-(6) are presented in Table 2 for the considered mono-crystalline PV module for STC (PV Future – PVF 60M) [31].

Table 2. Electrical parameters of the considered mono-crystalline PV module for STC.

  1. The electrical model is scarcely presented in subsection 2.2 but there is no explanation about how the parameters of such a model are adjusted. The reported model is implicit, so it requires solving techniques like numerical methods. If empirical IV curves have been used to adjust some of the parameters, for example the parasitic resistances, it should be clearly expounded.

Answer 16:

Thank you. Corrected.

In the paper, we focused more on presenting a novel thermo-electric model and discussing the obtained results. The detailed presentation of the double-diode model was not essential to us in this part, so the presentation in subchapter 2.2 was minimalist.

A numerical method called Trapezoidal Rule - Backward Difference Formula of the 2nd order (TR-BDF2) was used to solve the implicit equation of the double diode model (described in section 2.2.).

  1. In lines 281-282 the effect of degradation in power generation is slightly commented. In this regard, that statement could be enriched by mentioning some effects of degradation like cell cracking, yellowing, browning, etc.

Answer 17:

Thank you. Corrected.

Degradation of PV modules involves many types of effects such as cell cracking, yellowing, browning, and more [47,48]. The degradation rate takes effect when the output power of the PV module is reduced by at least 10 %. However, high-efficiency PV modules with a 50% degradation rate can be more efficient than low-efficiency PV modules with a lower degradation rate [49].

47.       Frick, A.; Makrides, G.; Schubert, M.; Schlecht, M.; Georghiou, G.E. Degradation Rate Location Dependency of Photovoltaic Systems. Energies 202013, 6751.

48.       Rajput, P.; Malvoni, M.; Manoj Kumar, N.; Sastry, O.S.; Jayakumar, A. Operational Performance and Degradation Influenced Life Cycle Environmental–Economic Metrics of mc-Si, a-Si and HIT Photovoltaic Arrays in Hot Semiarid Climates. Sustainability 202012, 1075.

49.       Dhimish, M.; Alrashidi, A. Photovoltaic Degradation Rate Affected by Different Weather Conditions: A Case Study Based on PV Systems in the UK and Australia. Electronics 20209, 650.

  1. The achieved results are validated on the view of error metrics (RMSE and MAE); however, there is no comparison with previous literature. Moreover, the common practice when modeling PV cells/modules under the electric equivalent circuit is obtaining the estimation of generated current, so the RMSE is obtained for such magnitude.

Answer 18:

Thank you. Corrected.

In the discussion section, we made an additional comparison between the presented results and the results of other studies. Thank you for the comment and advice in estimating the generated current instead of the output power. However, we decided to keep the current display for easier further comparison with other studies (which also show the results as the output power of the PV module).

  1. In addition, it is unclear the novelty or contribution of the reported research. The authors should emphasize the novelties and benefits in order to state the relevance and opportunity of their research. This must be mainly highlighted in the discussion or conclusions.

Answer 19:

Thank you. Corrected.

Based on other reviewers, section 3 “Results and discussion” was divided into individual sections. A discussion section was added where the results of the study were compared with other studies done so far. In both chapters (discussion and conclusion) we further highlighted the novelty and benefits of the research as you suggested.

Round 2

Reviewer 1 Report

The authors have conducted the requested changes.

The paper has been significantly improved.

It can be accepted for publication.

Reviewer 4 Report

The revised version of the manuscript addresses the provided suggestions in a proper manner. The quality and presentation of the paper has been enhanced. Congratulations to the authors for their efforts.